# Remote Sensing of Lake Sediment Core Particle Size Using Hyperspectral Image Analysis

**Hamid Ghanbari** [1,2,*] , **Olivier Jacques** [1] , **Marc-Élie Adaïmé** [1], **Irene Gregory-Eaves** [2,3] **and Dermot Antoniades** [1,2]

1. Department of Geography, Université Laval, Québec, QC G1V 0A6, Canada;
   olivier.jacques.7@ulaval.ca (O.J.); marc-elie.adaime.1@ulaval.ca (M.-É.A.);
   dermot.antoniades@cen.ulaval.ca (D.A.)
2. Groupe de Recherche Interuniversitaire en Limnologie, Université de Montréal, C.P. 6128,
   Succursale Centre-Ville, Montréal, QC H3C 3J7, Canada
3. Department of Biology, McGill University, Montréal, QC H3A 1B1, Canada; irene.gregory-eaves@mcgill.ca
* Correspondence: hamid.ghanbari.1@ulaval.ca

**Abstract:** Hyperspectral imaging has recently emerged in the geosciences as a technology that provides rapid, accurate, and high-resolution information from lake sediment cores. Here we introduce a new methodology to infer particle size distribution, an insightful proxy that tracks past changes in aquatic ecosystems and their catchments, from laboratory hyperspectral images of lake sediment cores. The proposed methodology includes data preparation, spectral preprocessing and transformation, variable selection, and model fitting. We evaluated random forest regression and other commonly used statistical methods to find the best model for particle size determination. We tested the performance of combinations of spectral transformation techniques, including absorbance, continuum removal, and first and second derivatives of the reflectance and absorbance, along with different regression models including partial least squares, multiple linear regression, principal component regression, and support vector regression, and evaluated the resulting root mean square error (RMSE), R-squared, and mean relative error (MRE). Our results show that a random forest regression model built on spectra absorbance significantly outperforms all other models. The new workflow demonstrated herein represents a much-improved method for generating inferences from hyperspectral imagery, which opens many new opportunities for advancing the study of sediment archives.

**Keywords:** hyperspectral image; random forest; mean particle size; sediment core; paleolimnology

## 1. Introduction

Hyperspectral imaging (imaging spectroscopy) has gained attention in the geoscience community because of its detailed representation of sediment features [1]. This technique can be used to determine the composition of sediment cores based on the analysis of absorptions at specific wavelengths of electromagnetic radiation [2]. Due to the rapid advance of scanning technologies and associated algorithms, hyperspectral imaging has emerged in recent years as a tool in paleolimnology, which is focused on tracking changes in a given lake as well as its watershed and airshed [3]. Hyperspectral imaging has shown great promise for the analysis of sediment cores with numerous possible applications [1,4,5], given that it is fast and non-destructive, and represents a high spatial resolution technique (i.e., submillimeter resolution). Butz et al. [4] modified the interpretation of relative absorption band depth (RABD) features, first introduced by Rein and Sirocko [6] for reflectance spectroscopy, and used them to infer algal chloropigments as well as bacterial pigments. In another study, Schneider et al. [7] developed a model for the calibration of sediment core chromatography to

reconstruct eutrophication history at an intra-annual timescale (i.e., within sediment varves). Sediment core hyperspectral imaging has also been applied to other questions such as documenting past fire regime changes by searching for charcoal in hyperspectral images [8], the prediction of organic matter content [9], investigation of phosphorus fraction retention variations [10], or detection of tephra layers in sediment cores [11]. However, corruption of the measured pixels by matrix effects (e.g., water content and porosity) may at times interfere with the interpretation of results [4].

Textural analysis of sediment cores (analysis of general physical characteristics), such as mean particle size (MPS), provides fundamental information for the study of the sedimentary record [12,13]. Particle size is widely used in sedimentological studies, as variations reflect changes in depositional processes, transport mechanisms, sediment sources, and erosional activities, and thus record information about a lake and its surrounding landscape. The different methods used for obtaining the MPS generally include direct measurements such as X-ray sedigraphy, gravimetric sedimentation, sieving, Coulter Counter analysis, and laser diffraction [14]. Although most of these techniques provide precise and reliable measurements for a particular range of particle sizes, they are destructive, time- and material-consuming, labor intensive, and low resolution [15]. Recently, visible near-infrared spectroscopy has been used to address some of these limitations, since it is rapid and operates without making physical contact with sediments [16–18]. Visible and near-infrared reflectance spectra of the sediments have shown potential to derive chemical and physical parameters, especially chlorophyll and particle size, at low spatial resolution [19,20].

Hyperspectral imaging has been applied to estimate particle size and has shown promising results in terms of precision and applicability. Jacq et al. [15] used integrated visible near-infrared (VNIR) and short-wave infrared (SWIR) hyperspectral images to estimate particle size distribution and fractions using partial least squares regression (PLSR) models. Although hyperspectral images show great potential for estimation of particle size and other textural characteristics, the choice of methodology and the use of appropriate multivariate analysis techniques are of great importance and have not yet been fully evaluated. Moreover, the preprocessing and transformation techniques applied for treating the raw data affects model development due to the expansive spectral range and the large volume of information contained in the spectral data [21]. The main goals of preprocessing are to optimize the spectral range, reduce the impact of external factors on the data, and highlight the pertinent spectral features, thus establishing the foundation of a rigorous predictive model [22]. In this study, we evaluated several approaches for inferring MPS, including support vector regression (SVR), partial least squares regression (PLSR), multiple linear regression (MLR), principal component regression (PCR), and random forest (RF) regression. In order to develop a robust and accurate predictive model, we created an integrated SWIR/VNIR hyperspectral image and subjected it to different preprocessing methods to analyze the effect of the spectral variables. We applied each method to six lake sediment cores for which measured MPS were available, and evaluated the models based on the root mean square error (RMSE), coefficient of determination (aka Nash Sutcliffe efficiency; subsequently referred to as $R^2$), and mean relative error (MRE) [23].

## 2. Material and Methods

### 2.1. Site Description and Coring

This study included 651 particle size samples collected from six Canadian lake sediment cores: five from southern Quebec and one from the Canadian Arctic (Table 1). The southern Quebec lakes studied are located south of the Saint Lawrence river in the Mixedwood Plains ecozone. These lakes are mesotrophic to hypereutrophic lakes that are situated in watersheds affected by agriculture, urbanization, and past mining activities. By contrast, Lake Fury 2 (unofficial name) is an oligotrophic lake situated in the Canadian Arctic; its catchment is undisturbed and sparsely vegetated. Sediment cores measuring 0.82–1.27 m in length were recovered from the lakes in summer 2017 using an Aquatic Research Instruments universal corer that recovered intact sediment-water interfaces from each lake.

## *2.2. Workflow Overview*

The first step in the methodology was the acquisition of visible-near infrared (VNIR) and shortwave infrared (SWIR) hyperspectral images (Figure 1, Step 1). After calibration and preprocessing of the VNIR and SWIR images (Step 2), they were registered together to constitute the integrated hyperspectral image (Step 3). By implementing a patch-wise strategy on the integrated hyperspectral image, the sample datasets were established (Step 4). Finally, after the selection of the most informative bands from the sample dataset (Step 5), different transformations and model development methods were applied and assessed (Step 6). Each step is described in detail in the following sections.

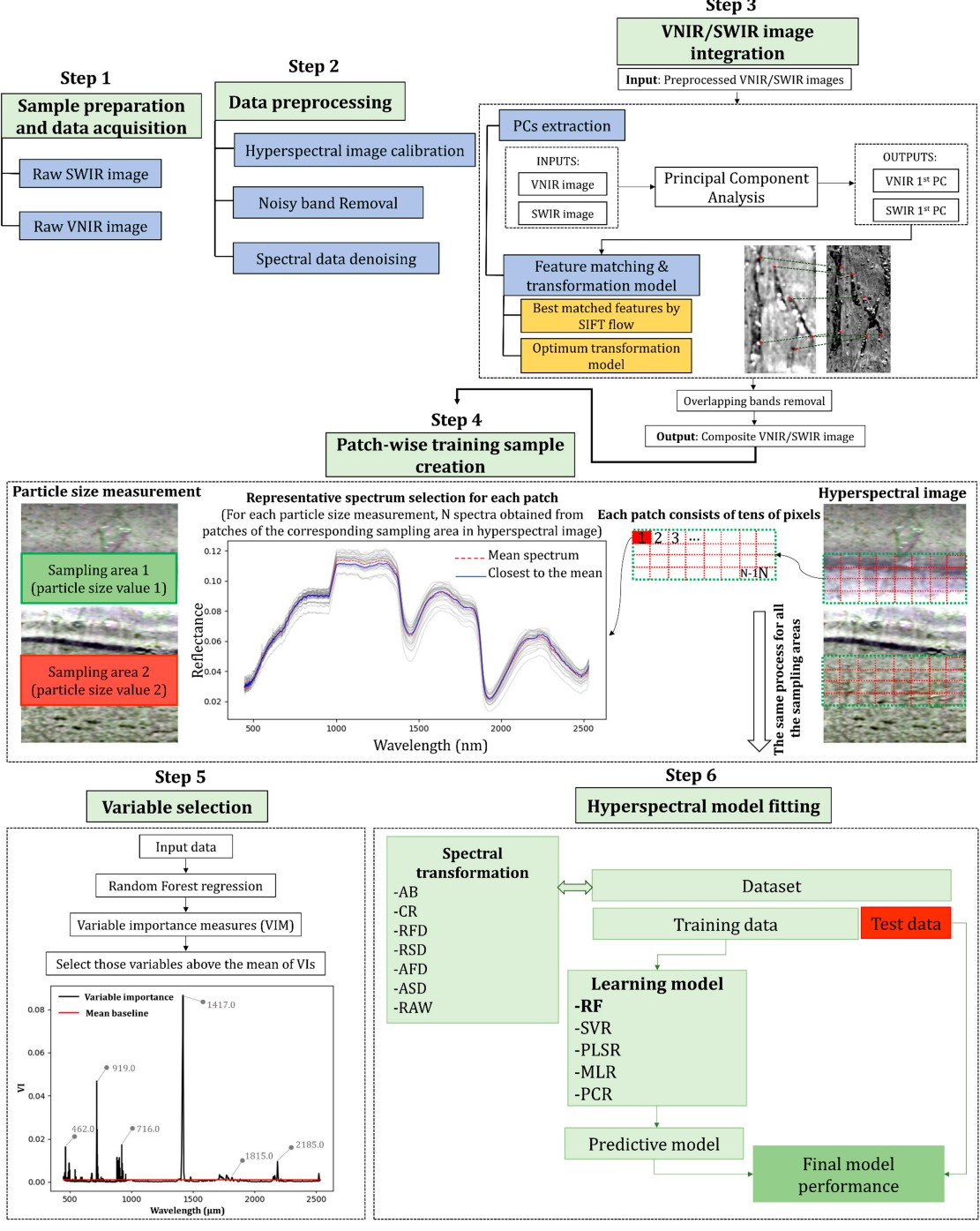

**Figure 1.** Schematic diagram of the proposed workflow for model development.

**Table 1.** Description of sampling sites.

| Lake Name | Code | Latitude | Longitude | Surface Area (km²) | Max Depth (m) | Core Length (m) | #Samples |
|---|---|---|---|---|---|---|---|
| William | WIL | 46°07′ N | 71°34′ W | 4.90 | 30.1 | 1.27 | 127 |
| Stater | STA | 46°04′ N | 71°28′ W | 0.36 | 3.9 | 1.14 | 114 |
| Bécancour | BEC | 46°04′ N | 71°14′ W | 0.97 | 3.4 | 1.13 | 113 |
| à la Truite | TRU | 46°05′ N | 71°30′ W | 1.24 | 2.5 | 1.10 | 110 |
| Joseph | JOS | 46°11′ N | 71°33′ W | 2.53 | 12.0 | 1.05 | 105 |
| Fury 2 | Fury | 69°39′ N | 82°33′ W | 0.11 | 20.0 | 0.82 | 82 |

## 2.3. Sample Preparation and Particle Size Measurement

The sediment cores were split lengthwise, with one half used for particle size and other destructive laboratory analyses; the other half was used for hyperspectral imaging and then preserved as an archive. Cores used for destructive analyses were sliced into discrete 0.5 or 1.0 cm sections and stored in sealed WhirlPak bags. Sample preparation and analysis of the size distribution of sediments was carried out with laser granulometry using a Horiba granulometer (model LA950v2) following the methods of Narancic et al. [24]. This device operates according to the Mie optical principle (including Fraunhofer approximation) for fine sediments whose size spectrum ranges from 0.11 to 3000 μm. The operation lasted approximately two minutes for each sample, from the introduction of the samples into the granulometer to the export of the results; mean particle size values were calculated based on the Folk and Ward [25] method using the Gradistat program [26].

## 2.4. Hyperspectral Data Acquisition

Prior to image analysis, the acquisition of high-quality sediment images is required, as low quality or substandard images will create significant problems during the subsequent analyses. Sediment core surfaces were cleaned and flattened by removing protrusions (using a knife, single-edge razor blade or a glass microscope slide) in order to get the optimum focus along the detector arrays and expose fine sediment structures, while removing irregularities such as depressions or bumps [27]. Because other planned analyses included light- and temperature-sensitive components such as pigments and DNA, leaving the cores to dry for hours was not possible, and so cores were covered with transparent plastic film to mitigate the effects of water reflectance. Each sediment core was positioned in the middle of the scanning table and in the field of view of the cameras, with the top of the core facing toward the focus grid and the camera, and the level of the core surface was adjusted to the same height as the focus grid. The scanning system uses two rows of seven 50-W quartz-halogen lamps to ensure optimum illumination angle and light intensity, with the light radiated to the surface of the sediment core in all directions (i.e., hemispherical incident radiance) and conical beam reflected back to the sensors [28].

After optimizing the operational parameters such as camera settings and scanning table speed, sediment cores were scanned with a scanner system equipped with two hyperspectral cameras (Specim Ltd., Oulu, Finland) working in the SWIR and VNIR wavelengths of the electromagnetic spectrum. The optical characteristics of the cameras are presented in Table 2.

**Table 2.** Optical characteristics of the hyperspectral cameras.

| Spectral Camera | PFD4K-65-V10E | Spectral Camera SWIR |
|---|---|---|
| Spectral range (nm) | 400–1000 | 1000–2500 |
| Spatial Resolution (pixel size) | ~40 μm | ~200 μm |
| Spectral sampling | 0.78–6.27 nm | 5.6 nm |
| Spectral bands | 776 | 288 |
| Radiometric Resolution (Bit) | 12 | 16 |

*2.5. Image Data Preprocessing*

2.5.1. Image Calibration

After capturing the hyperspectral data, the normalization of the raw hyperspectral image to the dark and white references was performed according to Equation (1):

$$HSI_{normalized} = \frac{HSI_{Raw} - DF_{av}}{WF_{av} - DF_{av}} \tag{1}$$

where $HSI_{Raw}$ is the Raw hyperspectral image, $DF_{av}$ is the dark reference, and $WF_{av}$ is the white reference averaged into a single frame. The output is the fraction of reflected light in relation to the white standard.

The analyses were performed on a selected transect in the middle of each sediment core. It is generally assumed that sediment is deposited relatively evenly on lake bottoms at the scale of sediment cores (i.e., typically with diameters <10 cm) and, therefore, that sediment layers are of approximately even thickness. Where present, visually evident laminations in our samples indicated that this was the case. We thus considered that core properties should vary little in cross-section and selected optimum lengthwise transects for two reasons. First, sediment surfaces are not homogenous and usually contain porosities and irregularities; therefore, undisturbed transects result in better model analyses. Second, computational time and memory usage are problematic issues which must be confronted (for 1 m of full sediment core, the data storage capacity exceeds 40 GB of memory).

2.5.2. Noisy Band Removal

Imaging spectroscopy is often corrupted by various types of noise, due mainly to internal sensor malfunction [29]. Hyperspectral sensors are usually affected by noise and have non-linear responses in different spectral bands [30]; one common source is noisy bands with low signal-to-noise ratios (SNR) [31]. Removing the bands that are susceptible to interference by unwanted noise is a crucial step to establishing a robust model between hyperspectral images and other proxies [4]. In this study, we characterized noisy bands by calculating the correlation coefficients between adjacent bands and removing those bands with correlation coefficients below 0.95 [30]. As such, 60 bands of the VNIR image, ranging from 395.82 to 440.58 nm, and eight bands of the SWIR image, ranging from 2538.16 to 2577.33 nm, were recognized as noisy bands and removed.

2.5.3. Spectral Data Denoising

Noisy spectral data can also be generated by external sources, such as high reflectance areas due to water (i.e., saturated regions) or weakly reflected regions because of high porosity, as well as by internal defects inside the instrument such as detector sensitivity or the heat produced by the sensor while it is operational [32]. In order to remove the interference of noise and highlight spectral features, several spectral preprocessing procedures have been proposed, including the standard normal variate [32], the detrend algorithm [33], and the autoscaling method [34]. To remove spectral noise and preserve spectral information, we adopted a three-degree polynomial Savitzky–Golay (SG) smoothing algorithm with window size of 11 points [35], following the recommendations of Vidal and Amigo [36] and Ru et al. [18].

*2.6. VNIR/SWIR Data Integration*

Data fusion techniques have been extensively employed on remote sensing images to fuse data from different sensors to improve model performance [37]. Recent studies showed that combining laboratory SWIR and VNIR images into one integrated image can provide complementary information and increase model accuracy [18,38]. In this study, the fusion of VNIR and SWIR images was performed using the scale-invariant feature transform (SIFT) flow algorithm introduced by Liu et al. [39]. In this

procedure, the VNIR image was down-sampled to 200 μm resolution to give it the same pixel size as the SWIR image. Subsequently, the registration was performed by subjecting the first principal components (PCs) of the VNIR and SWIR sensor images to the SIFT flow algorithm [40]. Finally, after transformation of the SWIR image to the VNIR image space and removing the overlapping bands between 969.29 and 1001.49 nm, an integrated hyperspectral image consisting of 932 bands was constructed.

## 2.7. Patch-Wise Calibration Sample Creation

To construct a model relating particle size assays to high-dimensional spectra from sampling image regions containing thousands of pixels (i.e., regions of interest; ROI), the patch-wise approach was used [41]. The purpose of the patch-wise strategy is to build different distributions of the spectral data and to make a high-population training set while incorporating both spectral and spatial information. In this procedure, ROIs were divided into patches of predetermined size and the closest spectrum to the mean spectra was selected as a representative spectrum for each patch. To compute the distance of spectra from the mean spectrum, the Euclidean distance of the cumulative spectrum, which was found to be the most suitable distance function for hyperspectral data, was used [42]. During the calibration process, each representative spectrum was associated with the equivalent measured particle size and treated as an independent sample (Figure 1, Step 4).

## 2.8. Variable Selection Using Random Forest

Random Forest (RF), first introduced by Breiman [43], is a variation of ensemble classification and regression trees, which consists of many decision trees each built over a random selection of calibration samples from the dataset (bootstrap aggregation) and a random subset of candidate variables [44]. With different tree structures and randomly selected variables, not every tree sees all the features or all the calibration samples, and this guarantees that the trees are not correlated and therefore less prone to over-fitting [45]. During the development of ensemble trees, two-thirds of the bootstrap samples, on average, are used to grow the bagged trees and the remaining samples (out-of-bag; OOB samples) are employed to independently cross-validate the model (see schematic in Figure 2).

A specific characteristic of RF is variable importance (VI), which permits not only evaluation and ranking of variables (i.e., wavelengths) in terms of relative significance but also interpretations of data and identification of the most prominent features [46,47]. There is a vast literature on variable selection using RF and it is still the subject of ongoing research [48,49]. A conservative strategy for RF variable selection is based on the changes of prediction accuracy in a recursive forward addition or backward elimination of variables until the best performance is achieved [48]. These approaches usually get acceptable results but require long computational times [49]. Selection can also be made by advanced but time-consuming approaches such as those of Genuer et al. [50], who introduced a two-step ascending variable selection procedure (i.e., VSURF) for accurate data interpretation and model prediction purposes. Here, to get a stable VI ranking, the RF model was run on all available samples for 25 iterations and the mean scores of importance (i.e., the average importance values of 25 runs) were considered as VI values. The variables of high importance (scores above the average of the VIs) were then selected for data interpretation and model development (see Figure 3).

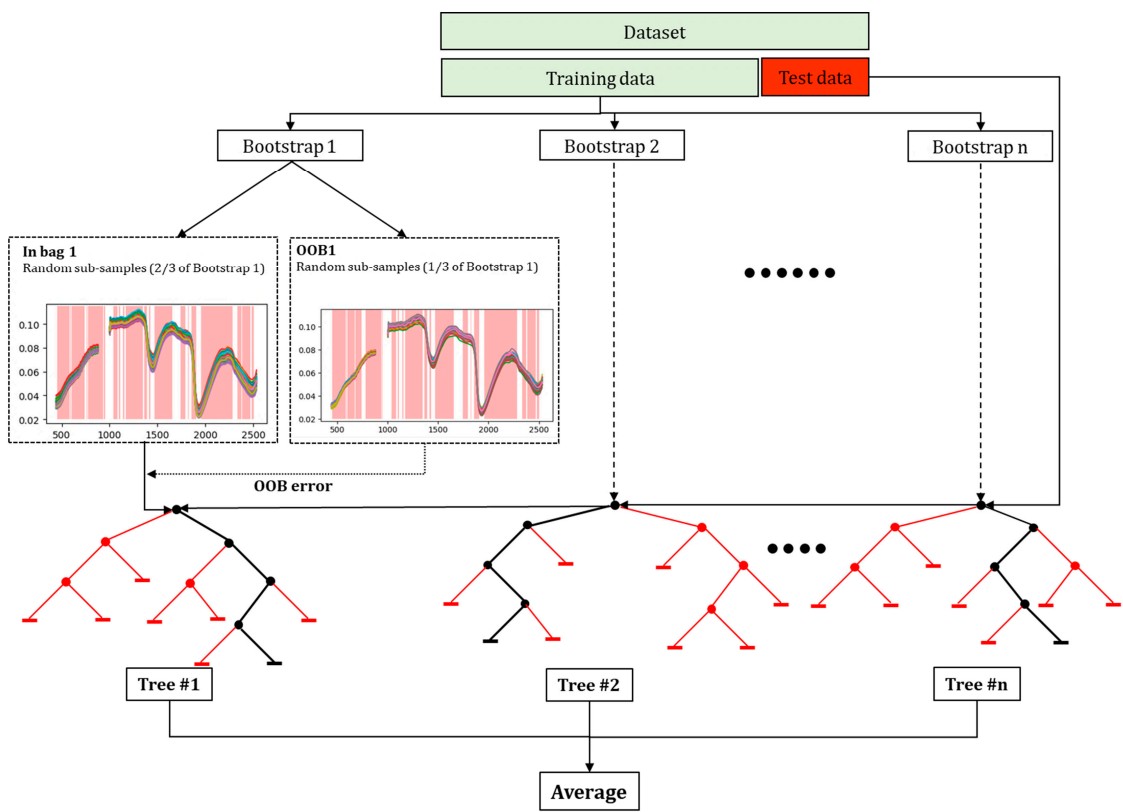

**Figure 2.** Schematic of the random forest regression model.

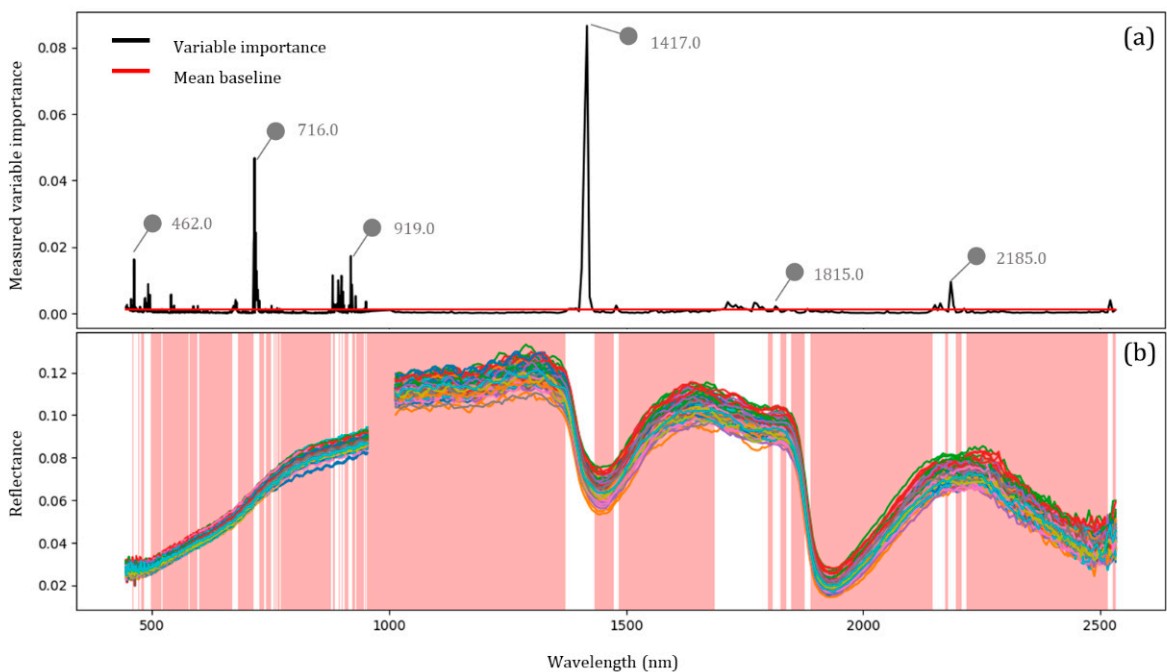

**Figure 3.** Variable importance values extracted by the Random Forest (RF) regression model (**a**) and removed variables based on the mean baseline (red region) (**b**).

*2.9. Hyperspectral Model Fitting*

Prior to fitting the models, we applied different transformation techniques to the raw spectral data to evaluate their performance in model development, including absorbance (AB), continuum

removal (CR), reflectance first derivative (RFD), reflectance second derivative (RSD), first derivative of absorbance (AFD), and second derivative of absorbance (ASD) transformations [51,52]. In the training process, the available input spectra were randomly divided into calibration (80%) and validation (20%) sets, and the calibration set was used for building the predictive models based on RF, SVR, PLSR, MLR, and PCR. The predictive models were generated and validated for the six lakes separately using their respective calibration and validation sets. An aggregate model was also generated based on a combined calibration set which incorporated all six individual datasets to examine the applicability of a single model across different lake environments.

Experiments were conducted using MATLAB and Python programming languages on a 3.4 GHz processor with 16 GB of RAM. To avoid bias, each scenario was repeated 10 times and the averages are reported. All RF regression models were constructed using the parameter values Mtry = n/3, nodesize = 5, and Ntree = 500 (n = number of bands). The grid search method was implemented to find the optimum SVR hyperparameters (kernel function = RBF and C = 1000). The partial least squares regression model was implemented with 10 orthogonal predictors.

## 3. Results

### 3.1. Evaluation of the Transformation Techniques

The results of the models developed using each of the datasets showed that the absorbance transformation (AB) was often substantially superior to other transformations in terms of prediction accuracy. In many cases, the other transformation techniques performed more poorly than analyses based on the raw spectra (Table 3). For most of the tested regression models, AB equaled or improved $R^2$ and RMSE when compared with other transformation techniques. In terms of MRE, AB had the lowest values in the best overall model (using random forest regression; discussed in Section 3.2), but in less performant models other transformation techniques had slightly better MREs.

### 3.2. Evaluation of the Predictive Models

The clear superiority of RF over other regression models is shown by the maximum $R^2$ values in both calibration and validation sets in all lakes (Figure 4). The RF models were stable and robust in calibration, as $R^2$ values were generally above 0.90. For the validation set, RF improved $R^2$ by approximately 21, 19, 11, 34, 9, and 24% over the second-best regression model (SVR) for WIL, JOS, TRU, BEC, STA, and Fury datasets, respectively (Figure 4b).

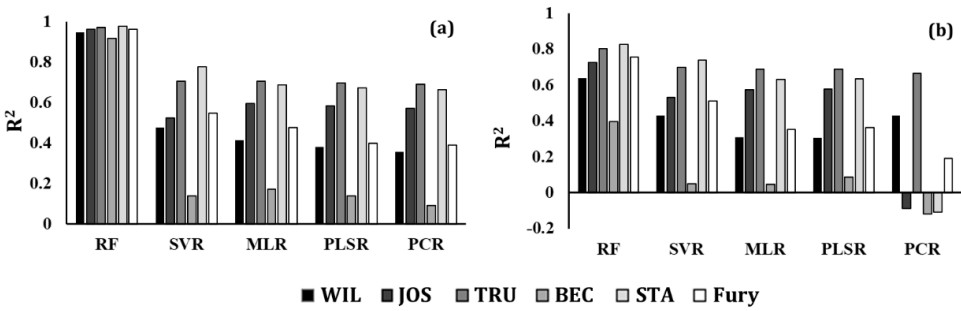

**Figure 4.** $R^2$ values of the individual predictive models on the AB-transformed data of the (**a**) calibration and (**b**) validation sets.

The high $R^2$ values that resulted from applying RF models to sediment core reconstructions show the great potential of these models to reproduce distinct particle size variations (Figure 5). Here, the $R^2$ values for each dataset are calculated between the average hyperspectral predictions of sampling areas and the measured MPS to validate the inferences of volume information of sediment cores from surface hyperspectral imaging. The comparison of predicted particle size time series with their corresponding measured values clearly shows strong coherence (Figure 5).

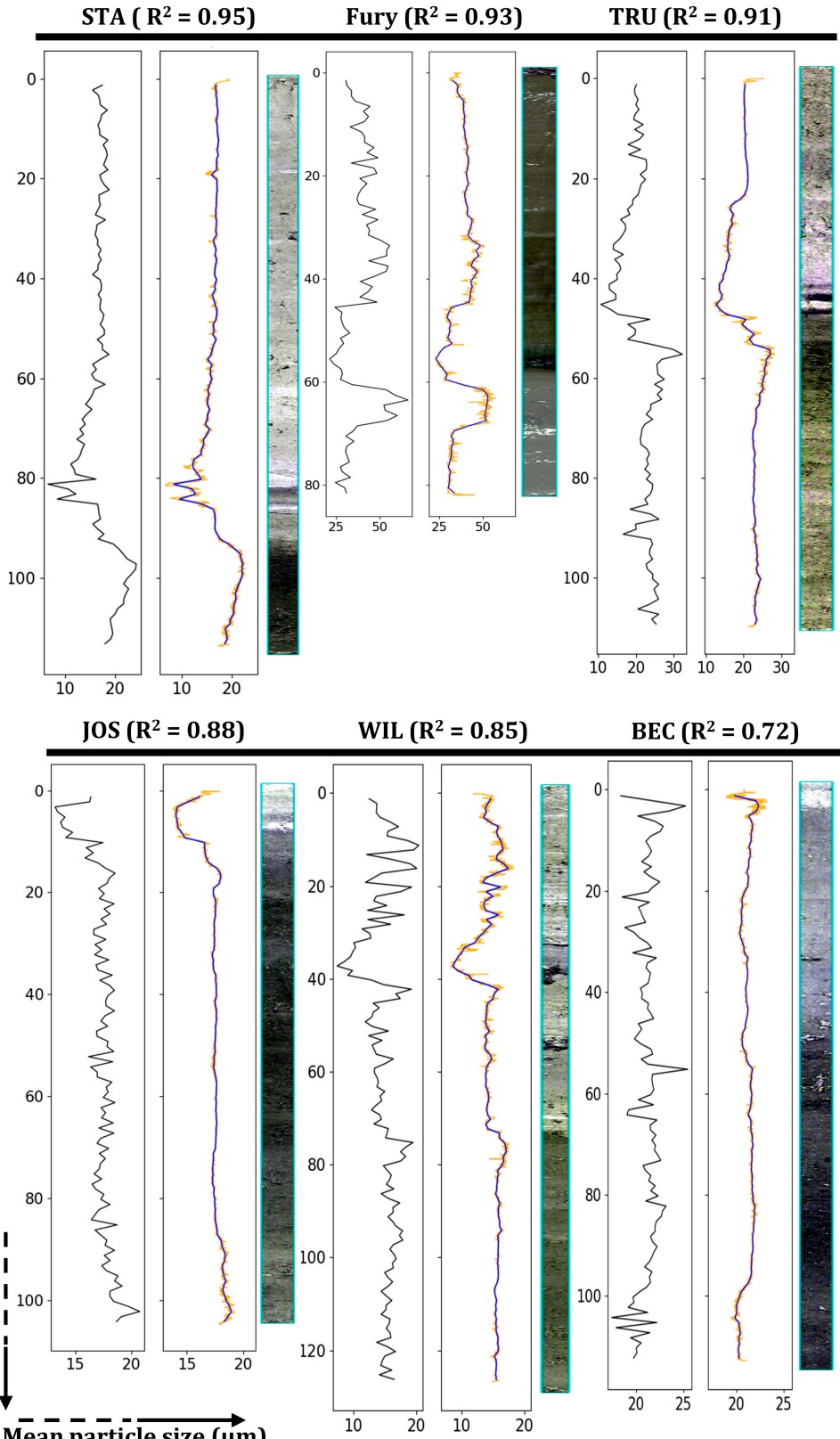

**Figure 5.** Measured mean particle size versus depth (left diagrams), and hyperspectral image-inferred values using the top-performing method (AB-RF method; right diagrams). Orange lines: high-resolution hyperspectral time series; blue lines: one-centimeter averages. At right are the enhanced RGB images of each sediment core.

**Table 3.** Comparative results for the different regression models and transformation techniques. Data are from the total set (calibration and validation). Values in bold indicate the best results for each model.

| Model | Transformation Technique | WIL | | | JOS | | | TRU | | | BEC | | | Fury | | | STA | | |
|---|---|---|---|---|---|---|---|---|---|---|---|---|---|---|---|---|---|---|---|
| | | $R^2$ | RMSE | MRE | $R^2$ | RMSE | MRE | $R^2$ | RMSE | MRE | $R^2$ | RMSE | MRE | $R^2$ | RMSE | MRE | $R^2$ | RMSE | MRE |
| RF | AB | **0.85** | **0.79** | **3.80** | **0.89** | **0.16** | **1.58** | **0.92** | **1.29** | 3.40 | **0.77** | **0.33** | **1.66** | **0.90** | 9.87 | 5.00 | **0.93** | **0.54** | 2.83 |
| | CR | 0.79 | 1.11 | 4.73 | 0.86 | 0.21 | 1.81 | 0.89 | 1.83 | 4.09 | 0.71 | 0.41 | 1.90 | 0.81 | 19.05 | 6.72 | 0.91 | 0.77 | 3.54 |
| | RFD | 0.75 | 1.34 | 5.29 | 0.82 | 0.26 | 2.04 | 0.81 | 3.04 | 5.36 | 0.66 | 0.49 | 2.11 | 0.78 | 21.95 | 7.67 | 0.88 | 0.99 | 4.17 |
| | RSD | 0.69 | 1.65 | 5.86 | 0.78 | 0.33 | 2.28 | 0.79 | 3.34 | 5.91 | 0.62 | 0.54 | 2.25 | 0.67 | 32.68 | 9.73 | 0.83 | 1.40 | 5.06 |
| | AFD | 0.73 | 1.47 | 5.44 | 0.79 | 0.31 | 2.22 | 0.82 | 2.80 | 5.21 | 0.64 | 0.51 | 2.18 | 0.74 | 25.78 | 8.45 | 0.86 | 1.16 | 4.58 |
| | ASD | 0.68 | 1.72 | 5.96 | 0.76 | 0.36 | 2.35 | 0.76 | 3.88 | 6.48 | 0.62 | 0.55 | 2.27 | 0.65 | 34.75 | 10.27 | 0.82 | 1.49 | 5.37 |
| | None | **0.85** | 0.81 | 3.82 | 0.88 | **0.16** | 1.61 | 0.92 | 1.30 | 3.42 | **0.77** | **0.33** | **1.66** | **0.90** | 9.89 | 5.02 | **0.93** | **0.54** | **2.80** |
| SVR | AB | **0.46** | **2.88** | **8.83** | **0.60** | 0.70 | 3.99 | **0.70** | **4.73** | 7.97 | **0.19** | **1.21** | 3.97 | **0.54** | **46.25** | **12.53** | **0.77** | **1.96** | **6.49** |
| | CR | 0.42 | 3.10 | 9.20 | 0.49 | 0.75 | 4.15 | 0.63 | 5.95 | 8.88 | 0.05 | 1.35 | 4.12 | 0.43 | 56.77 | 13.65 | 0.71 | 2.45 | 7.69 |
| | RFD | 0.05 | 5.11 | 12.52 | −0.07 | 1.59 | 5.61 | 0.10 | 14.40 | 16.05 | −0.19 | 1.70 | 4.84 | 0.00 | 99.82 | 21.34 | 0.17 | 7.00 | 12.04 |
| | RSD | 0.00 | 5.37 | 12.84 | −0.15 | 1.71 | 5.85 | 0.00 | 15.94 | 17.08 | −0.19 | 1.70 | 4.84 | −0.03 | 102.29 | 22.11 | 0.01 | 8.30 | 13.03 |
| | AFD | 0.23 | 4.13 | 10.71 | 0.25 | 1.11 | 4.68 | 0.53 | 7.56 | 10.58 | −0.17 | 1.66 | 4.79 | 0.02 | 97.71 | 18.05 | 0.51 | 4.11 | 10.35 |
| | ASD | 0.09 | 4.87 | 12.05 | 0.01 | 1.47 | 5.39 | 0.19 | 12.88 | 15.03 | −0.16 | 1.65 | 4.75 | 0.00 | 99.56 | 19.83 | 0.41 | 4.96 | 10.84 |
| | None | 0.36 | 3.43 | 9.60 | 0.58 | **0.64** | **3.67** | 0.69 | 4.95 | 8.08 | −0.17 | 1.66 | 4.81 | 0.22 | 77.42 | 16.22 | 0.67 | 2.79 | 8.33 |
| PLSR | AB | 0.36 | 3.43 | 15.45 | 0.60 | 0.62 | 6.45 | **0.71** | **4.85** | 20.14 | **0.16** | **1.20** | **4.34** | **0.47** | **50.16** | 25.77 | **0.71** | 2.84 | 19.04 |
| | CR | 0.35 | 3.46 | 15.40 | 0.48 | 0.77 | 6.44 | 0.59 | 6.59 | 20.69 | 0.09 | 1.29 | 4.45 | 0.33 | 67.05 | 25.22 | 0.64 | 3.07 | 19.01 |
| | RFD | 0.32 | 3.65 | 15.37 | 0.41 | 0.87 | 6.42 | 0.48 | 8.31 | 20.54 | 0.07 | 1.33 | 4.44 | 0.35 | 65.18 | 25.76 | 0.52 | 4.04 | 18.48 |
| | RSD | 0.23 | 4.09 | 15.01 | 0.37 | 0.93 | 6.33 | 0.44 | 8.89 | 20.36 | 0.06 | 1.35 | 4.41 | 0.16 | 83.50 | 24.42 | 0.45 | 4.61 | **17.84** |
| | AFD | 0.26 | 3.95 | 15.24 | 0.37 | 0.93 | 6.34 | 0.56 | 6.96 | 20.92 | 0.07 | 1.33 | 4.44 | 0.29 | 70.37 | 25.26 | 0.55 | 3.79 | 18.76 |
| | ASD | 0.20 | 4.29 | **14.72** | 0.25 | 1.10 | **6.16** | 0.44 | 8.89 | 20.29 | 0.05 | 1.35 | 4.40 | 0.17 | 82.78 | **24.32** | 0.48 | 4.37 | 17.94 |
| | None | **0.42** | **3.16** | 15.49 | **0.61** | **0.54** | 6.41 | 0.69 | 4.88 | 21.00 | 0.12 | 1.26 | 4.52 | 0.46 | 54.40 | 26.09 | 0.69 | **2.61** | 19.00 |
| MLR | AB | 0.38 | 3.31 | 15.65 | **0.61** | 0.61 | 6.49 | **0.70** | **4.78** | 20.21 | **0.14** | **1.23** | 4.63 | 0.44 | 55.94 | 26.38 | 0.67 | 2.75 | 19.11 |
| | CR | 0.39 | 3.28 | 15.62 | 0.52 | 0.70 | 6.53 | 0.61 | 6.15 | 20.84 | 0.12 | 1.26 | 4.57 | 0.41 | 58.88 | 26.10 | 0.66 | 2.88 | 19.16 |
| | RFD | 0.31 | 3.68 | 15.44 | 0.42 | 0.86 | 6.43 | 0.41 | 9.34 | 21.07 | −0.17 | 1.66 | 5.05 | 0.38 | 62.26 | 26.19 | 0.53 | 3.99 | 18.53 |
| | RSD | 0.24 | 4.07 | 15.01 | 0.37 | 0.93 | 6.38 | 0.45 | 8.81 | 20.35 | −0.02 | 1.45 | 4.60 | 0.17 | 82.93 | 24.49 | 0.46 | 4.58 | **17.86** |
| | AFD | 0.26 | 3.94 | 15.25 | 0.37 | 0.93 | 6.36 | 0.57 | 6.88 | 20.94 | 0.07 | 1.33 | 4.45 | 0.30 | 69.73 | 25.38 | 0.56 | 3.74 | 18.80 |
| | ASD | 0.20 | 4.28 | **14.72** | 0.26 | 1.10 | 6.18 | 0.44 | 8.95 | 20.41 | 0.05 | 1.36 | **4.40** | 0.17 | 82.57 | **24.37** | 0.48 | 4.36 | 17.94 |
| | None | **0.44** | **3.02** | 15.74 | **0.61** | 0.58 | 6.45 | 0.70 | 4.82 | 21.06 | 0.13 | 1.24 | 4.61 | **0.51** | 49.08 | 26.73 | **0.70** | **2.55** | 19.08 |
| PCR | AB | **0.33** | **3.54** | 15.37 | **0.56** | 0.56 | **6.40** | 0.69 | 4.94 | 21.12 | **0.05** | 1.35 | 4.43 | **0.37** | **62.64** | 25.56 | **0.51** | 4.64 | 19.36 |
| | CR | 0.28 | 3.83 | 15.20 | 0.41 | 0.87 | 6.47 | 0.55 | 7.14 | 20.71 | 0.04 | 1.36 | 4.35 | −0.27 | 126.32 | 25.24 | 0.51 | **4.09** | 18.92 |
| | RFD | 0.23 | 4.10 | 15.08 | 0.24 | 1.13 | 6.11 | 0.39 | 9.68 | 20.23 | 0.02 | 1.40 | 4.28 | −0.28 | 127.81 | 24.57 | 0.41 | 4.95 | 17.61 |
| | RSD | 0.09 | 4.85 | **14.16** | 0.21 | 1.18 | 5.94 | 0.19 | 12.82 | 19.51 | 0.02 | 1.40 | 4.27 | −0.04 | 104.15 | **22.79** | 0.31 | 5.82 | **17.15** |
| | AFD | 0.21 | 4.25 | 14.90 | 0.27 | 1.09 | 6.16 | 0.53 | 7.56 | 20.81 | 0.01 | 1.41 | 4.27 | 0.14 | 85.48 | 24.27 | 0.47 | 4.46 | 17.91 |
| | ASD | 0.11 | 4.74 | **14.17** | 0.19 | 1.21 | 5.96 | 0.23 | 12.28 | **19.65** | 0.02 | 1.40 | **4.26** | −0.07 | 106.56 | 23.28 | 0.42 | 4.88 | 17.42 |
| | None | **0.33** | 3.61 | 15.43 | **0.56** | 0.64 | 6.45 | 0.68 | 5.15 | 20.98 | 0.04 | 1.37 | 4.41 | −0.39 | 138.77 | 25.88 | 0.45 | 4.64 | 19.74 |

### 3.3. Evaluation of the Variable Selection Method

Pairwise comparisons of $R^2$ values of the three best regression models (RF, SVR, and PLSR) show a good agreement between the models built using the full spectra and selected wavelengths (Figure 6). Most points are near the 1:1 line in both the calibration and validation scatter plots, indicating similar behavior for the full and selected wavelength models. However, in addition to lower $R^2$ values, the PLSR and SVR models also had a number of points more divergent from the 1:1 line, suggesting that with these models, stronger performance would be achieved using the full spectral range. By contrast, for RF regression, the performance of the models using selected wavelengths was nearly equivalent to those developed using the full spectral range. The comparison of processing times between the two scenarios (with or without variable selection) clearly shows that using selected wavelengths alone noticeably decreased the computational time, on average by 5, 7 and 8.5 times for SVR, RF, and PLSR, respectively.

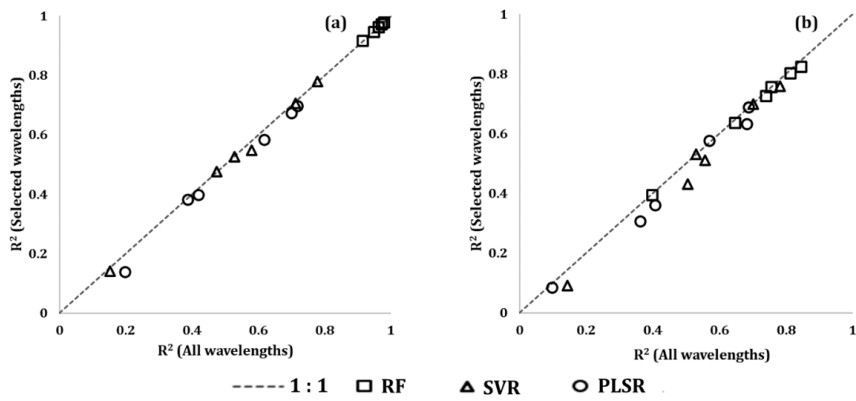

**Figure 6.** Comparison of $R^2$ values for the full and selected wavelength models for the (**a**) calibration set and (**b**) validation set.

### 3.4. Evaluation of the Aggregate Model

It is widely acknowledged that models based on individual datasets more accurately characterize material properties [53]. To be broadly applicable, however, a model should perform well in a variety of different systems. Therefore, after demonstrating the efficacy of the methodology we constructed an aggregate model—integrating all available particle size samples rather than those of individual cores—in order to test the performance and applicability for MPS prediction across a variety of lake systems (Figure 7, Table 4). Individual models outperformed the aggregate model in their respective lakes, but the aggregate model was still reasonably accurate, with an average $R^2$ of 0.81 across all cores (Table 4). Although the reported $R^2$ showed an average decline of 0.07 units for the aggregate model, we found that the predictive strength of the aggregate model was stable among the different datasets. While the applicability of the aggregate model may be related to different particle sizes (discussed in Section 4), high $R^2$ and low RMSE and MRE values confirm that aggregate model is highly reliable (Figure 7).

**Table 4.** Comparative results for the individual and aggregate RF regression models.

| Dataset | Individual RF Models | | | Aggregate RF Model | | |
|---|---|---|---|---|---|---|
| | $R^2$ | RMSE | MRE | $R^2$ | RMSE | MRE |
| STA | 0.95 | 1.01 | 4.87 | 0.92 | 1.13 | 5.52 |
| Fury | 0.93 | 4.29 | 8.24 | 0.92 | 4.74 | 8.63 |
| TRU | 0.91 | 1.66 | 5.85 | 0.87 | 2.27 | 7.69 |
| JOS | 0.88 | 0.59 | 2.81 | 0.79 | 0.78 | 3.61 |
| WIL | 0.85 | 1.30 | 6.68 | 0.75 | 1.67 | 9.41 |
| BEC | 0.72 | 0.90 | 2.94 | 0.60 | 0.98 | 3.07 |

## 4. Discussion

The use of hyperspectral images for quantifying MPS requires advanced multivariate analysis techniques, which differentiate the response of the different particle size features through spectral characteristics [46]. In paleolimnology, most existing studies that have developed models for specific materials have given emphasis to the MLR, PLSR, and SVR methods [9,54,55]. However, with the growth of spectroscopy applications and the diversity of sediment characteristics from different catchments, the need for more stable and accurate predictive models is increasing. Based on the accuracy metrics for the total sets, and the $R^2$ values of the calibration and validation sets, the RF algorithm produced predictive models with higher accuracy and stability compared to other methods, followed by the SVR and PLSR models (Table 3; Figure 4). Moreover, among different transformation techniques, AB transformation-based models revealed a performance superior or equivalent to those models developed using the raw spectral data. In some cases, the behavior of models such as PCR was not consistent among different transformation techniques, and some negative $R^2$ values for PCR were reported. As $R^2$ compares the fit of the model with that of a horizontal straight line, negative values happen only when the model does not follow the trend of the data and so fits worse than a horizontal line. Given the superiority of the models, the AB-RF method was applied to the sediment core hyperspectral images to infer mean particle size, which was then compared with the low-resolution time series measured with granulometry (Figure 5). The high $R^2$ values for the six datasets (all ≥ 0.72, and 3 of 6 cores ≥ 0.90) together with the highly similar curves, indicate that our proposed hyperspectral image-based methodology can replicate more time-intensive laboratory analyses with high reliability.

Hyperspectral imaging of sediment cores captures information in hundreds of bands, providing the great advantage of building more accurate models using only the most relevant data. Variable importance metrics and the related selected bands (Figure 3) are computed based on the contribution of each band to model development, and they enable the identification of spectral regions that have the greatest impact on the calibration model. In our case, the positions of these key regions are related to the spectral responses of different materials in specific parts of the electromagnetic spectrum [56]. The broad selected variables at wavelengths below 1000 nm may result from iron oxides, carotenoids, and chlorophyll pigments [15,57]. Narrow, well-defined selected bands near 1400 and 1900 nm are mainly associated with the O-H bonds of residual water in organic matter, and the features near 2200 nm may arise from overtones of vibrations related to organic matter or clay minerals [58,59]. However, lake sediments are complex mixtures of different organic and inorganic components, and although the variable selection method efficiently detects those wavelengths that are most strongly associated with changes in MPS, it lacks the capacity to determine the exact characteristics underlying these variations. Comparing different models constructed using all hyperspectral image bands to those with wavelengths selected by the variable importance metric, there was little evident variation in model performance. For SVR and PLSR regression models, accuracies were higher when all bands were used for model fitting because of the greater number of features contributing to the fitting process. However, the results of the RF model using selected wavelengths and full spectra showed almost equivalent accuracies. Therefore, we conclude that the RF variable selection method maintains model accuracy, in addition to the advantages of enabling interpretation and lowering computational time requirements.

The stability of the aggregate model across datasets can be seen for different particle sizes, although the aggregate model appears to underestimate particle sizes above ~40 μm, likely due to the limited number of training samples in this size range (Figure 7a,b). As an example, Figure 7c shows the variation of the prediction error with respect to predicted particle size for the Fury dataset, which contains particle sizes above ~40 μm. The prediction error increases in regions of the image with larger particle sizes. Overall, the concentrations of the points distributed along the 1:1 line and the values of $R^2$, RMSE, and MRE indicate the generally strong concordance between predicted and measured particle size. However, prediction accuracies of the model at particle sizes greater than ~40 μm may require further investigation, including assessments of the stability of the RF model following the inclusion of samples that span a wider range of particle sizes. At the other end of the size range,

cores with very low values and limited particle size variations, such as BEC (overall range 18–25 μm), may return lower coefficients of determination because prediction errors represent a proportionally larger part of the overall predicted variance. However, even with lower $R^2$, the models remain highly accurate when evaluated by RMSE and MRE. We also recommend that the RF model be compared with other particle size measurement techniques, which tend to be designed to make measurements only within a specific range of particle sizes. For example, sieving is a technique where particles greater than 63 μm diameter are commonly lumped together, whereas pipette analysis is only suitable for the analysis of particles less than 63 μm diameter [14].

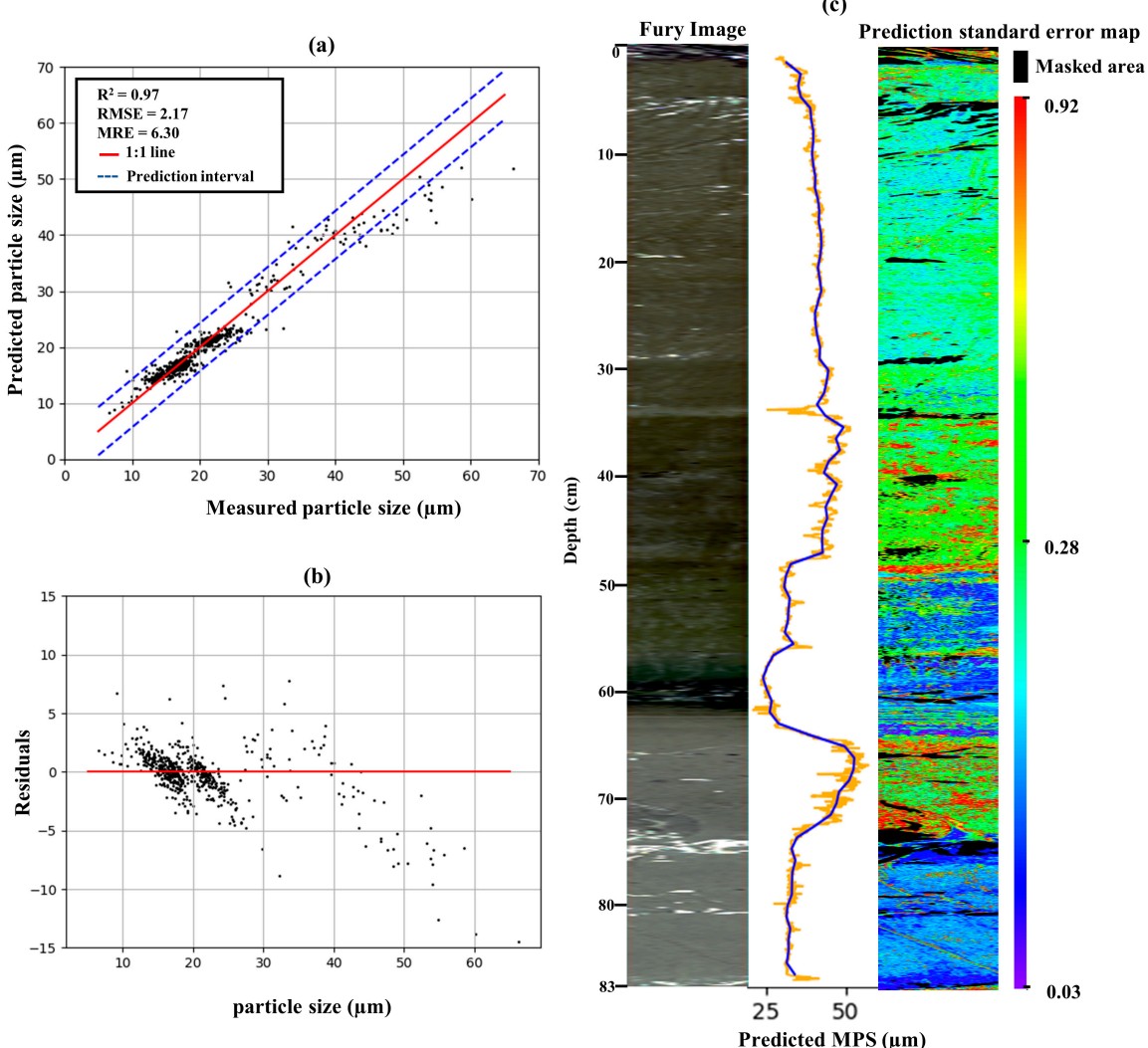

**Figure 7.** Ensemble scatter plots of (**a**) measured versus predicted particle size, and (**b**) residuals of the predicted particle sizes. (**c**) Graphical representation of predicted mean particle size (MPS) for Fury dataset and equivalent prediction standard error map (middle diagrams; orange line: high-resolution MPS profile; blue line: one-centimeter averages).

## 5. Conclusions

In this paper, we developed a hyperspectral image-based methodology to deal with the complexity of MPS determination from the information in the VNIR and SWIR portions of the electromagnetic spectrum. We evaluated numerous combinations of transformation techniques and regression models. Based on our results we conclude that:

1. Among the studied spectral variables, the AB transformation technique produced the best results. It is noteworthy that in some cases the raw spectra (without transformation) performed equally well or improved accuracy compared to AB.

2. The experimental results on the six datasets showed that the RF regression model outperformed all other regression models, with significant improvements relative to the second-best model (SVR). The stability and accuracy of the models constructed with different methods followed the order RF>SVR>PLSR>MLR>PCR.

3. RF variable selection using variable importance metrics reduced the computational burden and increased the interpretability of the data presented, at no cost to the strength of the models. Results from regression models with and without variable selection showed highly similar results.

4. Hyperspectral image-inferred mean particle size distributions in six sediment cores closely reproduced values measured with granulometry, confirming the utility of the technique for rapid, high-resolution environmental reconstructions. The performance of the aggregate model established its reliability and applicability to sediment cores from a variety of aquatic environments.

**Author Contributions:** Conceptualization, D.A. and H.G.; methodology and formal analysis, H.G.; data curation, O.J. and M.-É.A.; writing—original draft preparation, H.G. and D.A.; writing—review and editing, all authors; funding acquisition, D.A. and I.G.-E. All authors have read and agreed to the published version of the manuscript.

**Funding:** This work was funded by the Fonds de recherche du Québec–Nature et technologies (FRQNT), the Groupe de Recherche Interuniversitaire en Limnologie strategic network, the Canada Foundation for Innovation, and the Natural Sciences and Engineering Research Council of Canada (NSERC). The APC was funded by NSERC.

**Acknowledgments:** We are grateful to Saeid Homayouni for helpful comments on the manuscript. We are grateful to three anonymous reviewers for providing constructive reviews of the manuscript.

**Conflicts of Interest:** The authors declare no conflict of interest.

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
