# Peer review of "Remote Sensing of Lake Sediment Core Particle Size Using Hyperspectral Image Analysis"

_remotesensing, doi:10.3390/rs12233850_

Round 1
Reviewer 1 Report
This study introduced a new methodology to infer particle size distribution from laboratory hyperspectral images of lake sediment cores. This manuscript is well organized and interesting. I think it can be accepted by RS.
Author Response
We thank the reviewer for their time and appreciate their positive comment on the manuscript.
Reviewer 2 Report
This well-written manuscript presents a thorough analysis of numerical methods to be applied to sediment core spectral imagery for rapid and non-invasive particle size retrieval. The authors showed a clear advantage of using random forest approach for such a purpose leading to reasonable performances for sediment particle (mean?) size smaller than ~40 µm. The article is of interest for the geoscience community but the title is a little misleading since it seems to mention satellite or field sensors activities but it dedicated to laboratory methods instead. I would advise to slightly change the title to make it clearer (e.g., add “sediment core”). The methodology and approach are thoroughly described and performances nicely quantified. Nevertheless, it would be of interest to rely such techniques and consecutive evaluation on the physical explanation on how size distribution modifies spectral reflectance measurements. For instance, the authors could give some insights on how changes in PSD affect the reflectance signal for a given sediment type (same refractive index, similar shapes). Likewise, the PSD parameters are not rigorously defined: is it the mean, modal, median radius in number, surface or volume distribution?
Those two points have to be clarified before publication. I suggest to expand the discussion section on the physical information content and potential limitation for particles coarser than 40 µm as well as assessment of multimodal size distribution. Another point on the conclusions: the authors could advocate to directly use of raw data since results are statistically similar to those obtained after spectral transformation and would be much easier to apply.
Provided supplementary information on the physics of the problem and consideration of the minor comment, I would suggest publication of the manuscript in Remote Sensing.
Minor comments:
Title: specify ‘sediment core’
Figure 1: define acronyms for spectral transformation and learning model in caption. Add ‘Hyperspectral Image (HIS)’ in Step 2.
Section 2.4: I could not find any mention/description of the illumination setup. What lamp, illumination (and viewing) angles…
L.127: “transparent plastic film to mitigate the effects of water”: do you have any reference for this. It seems that you could measure the plastic signal without sediment as blank measurements and then subtract it to the actual measurements. Another point on the presence of water: water is extremely absorbing in the SWIR, how this perturbs data acquisition and processing.
L.142: can you specify what type of reflectance you obtain (see Schaepman-Strub, G., Schaepman, M. E., Painter, T. H., Dangel, S., & Martonchik, J. V. (2006). Reflectance quantities in optical remote sensing-definitions and case studies. Remote Sensing of Environment, 103(1), 27–42. https://doi.org/10.1016/j.rse.2006.03.002)
L.184: Why do you remove the overlapping bands; those bands could be used to assess good quality of fusion.
L.220 could you define what you called “mean scores”
L.233: please add a short notice/description on those different methods
L.245: it is not very clear to me that absorbance transformation outperforms the simpler use of raw data
Table 3. Specify the definition used for R2 showing negative values
Discussion: add section on impact of PSD on spectral reflectance. And discuss your findings for multimodal sediment size distribution
Reviewer 3 Report
The article presents an interesting methodological study related to the application of hyperspectral images to determine lake sediment particle size distribution. The article would be improved with a more detailed presentation of the results.
Notes to the manuscript
It is necessary to present more details about the analyzed samples - how 651 samples were obtained, how many samples are from each lake, what are the results of the measurement of PSD.
Formula for mean relative error MRE is not provided.
Are the same number of factors included in the regression equations of a given type when using different methods for transforming the spectral data?
The results of the validation sets are not presented in the manuscript. Only partial results are given in Figure 4.
Using only R2 to compare models, obtained for groups with different ranges of values and SD, can lead to incorrect conclusions, because value of R2 depends on these parameters.
Figure 5 - the gray and black lines of the graph are not clearly distinguished.
Table 4 - The RMSЕ values for BEC lake for the individual model and for the aggregate model are approximately the same – 1.01 and 0.98, but the MRE values are very different – 4.83 and 3.07, respectively.
What is the explanation for the difference in the accuracy of the models for the samples from the different lakes? For example, all models for lake BEC have very low accuracy.
Round 2
Reviewer 2 Report
The authors nicely replied to all my comments and criticisms. I consider now that the manuscript is in good shape for publication.